# Numeracy skills learning of children in Africa:—Are disabled children lagging behind?

**Huafeng Zhang** [1,2] *, **Stein T. Holden** [1]

**1** School of Economics and Business, Norwegian University of Life Sciences, Ås, Norway, **2** Fafo Institute for Labour and Social Research, Oslo, Norway

* zhu@fafo.no

**Data Availability Statement:** The data is openly available for all at: https://mics.unicef.org/surveys.

**Funding:** The paper has been undertaken as part of the research project "Education outcome variability in children with disabilities: Structure, institution or

## Abstract

Significant progress has been achieved in universal basic education in African countries since the late 1990s. This study provides empirical evidence on the within- and across-country variation in numeracy skills performance among children based on nationally representative data from eight African countries (DR Congo, The Gambia, Ghana, Lesotho, Sierra Leone, Togo, Tunisia, and Zimbabwe). We assess whether and to what extent children with disabilities lag in numeracy skills and how much it depends on their type of disabilities. More specifically, we explore whether disabled children benefit equally from better school system quality. The assessment is analysed as a natural experiment using the performance of non-disabled children as a benchmark and considering the different types of disabilities as random treatments. We first evaluate the variation in average numeracy skills in the eight African countries. They can roughly be divided into low- and high-numeracy countries. We apply Instrumental Variable (IV) methods to control the endogeneity of completed school years when assessing subjects' school performance and heterogeneous disability effects. Children with vision and hearing disabilities are not especially challenged in numeracy skills performance. The low numeracy skills among physically and intellectually disabled children are mainly attributable to their limited school attendance. Children with multiple disabilities are constrained both by low school attendance and by poor numeracy skills return to schooling. The average differences in school performance across the high- versus low-numeracy skill country groups are larger than the within-group average differences for disabled versus non-disabled kids. This indicates that school enrolment and quality are crucial for children's learning of numeracy skills, and that disabled children benefit equally from better school quality across these African countries.

## 1. Introduction

The Sustainable Development Goal (SDG) 4 aims at inclusive and equal access to education for all children [1]. Significant progress has been achieved [2] since the adoption of several development frameworks, such as Education for all [3] and the Millennium Development Goals [4]. Data from UNESCO Institute for Statistics suggests that since the late 1990s and

agency?" funded by the Research Council of Norway (Project Number 300635). The funders had no role in study design, data collection and analysis, decision to publish, or preparation of the manuscript.

**Competing interests:** The authors have declared that no competing interests exist.

early 2000s, most African countries have increased the gross enrolment in primary schools from about 75% to almost 100%. Even countries with low school enrolment historically, such as Niger, also witnessed their primary school gross enrolment to grow from 30% in the late 1990s to about 60–70% in recent years [5].

Although universal basic education has achieved great success, recent studies are concerned about poor school performance among children across African countries [6]. Many children did not gain basic skills in reading and mathematics even after many years of schooling [6–8]. Furthermore, the achievement gained in school enrolment has masked problems related to unequal distribution and disparity in school performance, as well as the marginalisation of the most disadvantaged and vulnerable groups of children [9–12]. Children with disabilities are possibly among those exposed to such limitations and risks [13, 14].

This paper aims to investigate the learning outcome in form of numeracy skills for children with and without disabilities in eight African countries. Based on the sixth round of Multiple Indicator Cluster Surveys (MICS), we aim to answer the following research questions: 1) To what extent do the average numeracy skills vary across African countries? 2) To what extent does the average performance differ between children with and without disabilities? 3) To what extent is the numeracy skills return to schooling dependent on children's disability status and types of disabilities? 4) Are disabled children able to benefit from better school system quality to the same extent as non-disabled children? To answer these questions, we first evaluate the variation in the numeracy skills across the eight African countries in our study and estimate the disability effects on numeracy skills returns to schooling by using non-disabled children as the counterfactual. Afterwards, we assess the relative performance between disabled and non-disabled children in countries with low- and high-numeracy skills. The country-level school quality is measured by the mean numeracy score of non-disabled children in these countries.

There is a growing research interest in timely and reliable empirical evidence on school enrolment and learning performance for children with disabilities in developing countries to measure the across-region variation [15, 16]. Several comparative studies based on data from multiple countries provided evidence on disabled children's overall low school attendance, enrolment, and school completion [15, 17–21]. However, none of these comparative studies has assessed disabled children's school performance.

Earlier studies based on Western experiences have presented evidence for learning challenges among disabled children since they are often limited in cognitive, behavioural, motor, and emotional abilities [22, 23]. However, the evidence on disability gaps in learning skills in the developing context is limited and primarily through simple tests embedded in surveys in individual countries. For example, studies in India [24] and Pakistan [25, 26] suggest a significant disability gap in numeracy skills. These studies do not indicate whether the low numeracy skills among disabled children have been merely correlated with their low school attendance or have originated from their challenges in learning in school. Takeda and Lamichhane (2018) notice that when the interaction between disability status and school status is included in the model, the disability dummy becomes insignificant [24]. They conclude that once disabled children access education, they become less likely to fall behind in school performance. We suggest that the endogeneity of selection into schooling should be considered when estimating the disability gap in learning.

Due to the challenges in sample size for disabled children, many studies in the developing context used the catch-all category for disability. There are a few exceptions. Singal et al. (2018) evaluated Pakistani children's basic numeracy skills among those with three types of disabilities and varying severity: sensory (walking, seeing and hearing); self-caring (difficulty in dressing and washing all over); and cognitive [26]. They only found significant disability

gap in learning outcomes among those with moderate or severe sensory disabilities but not among those with mild disabilities or other disability types.

Another study that also differentiates disability types is Bakhshi, Babulal and Trani (2018), who predicted school access and literacy in Western Darfur in Sudan for children with four types of disabilities: sensory (physical, seeing and hearing), mental and cognitive, behavioural and mood, and multiple disabilities [6]. They found no difference in skills performance either with the catch-all disability category or with different disability types. However, the authors further argue that in the conflict setting in Darfur, where all children are exposed to a high risk of being excluded and not taught in school, the differences in school performance might have disappeared. More evidence is needed to understand the heterogeneous effects of disability types and the potential correlation between the disability effect and school quality.

To the best of our knowledge, our study is the first comprehensive study evaluating disabled children's achievement in numeracy skills based on the standardised WG-CFM and numeracy tests in African countries. Our analysis uses the natural experiment framework by using the sample of non-disabled children as a benchmark (counterfactual). When assessing subjects' numeracy skills returns to schooling and heterogeneous disability effects, we apply Instrumental Variable (IV) methods to control the endogeneity of completed school years, since the disability status may directly affect children's likelihood of being in school.

## 2. Conceptual framework

This paper explores whether children with disabilities lag in numeracy skills compared to non-disabled children and to what extent such a lag varies with their disability status, school enrolment and country-level numeracy performance. Children with disabilities might lag in numeracy skills if they do not attend school as much as their non-disabled peers. Earlier studies found that children with disabilities are exposed to a higher risk of not attending school, enrolling late, or dropping out of school early [15, 17, 18, 27]. The factors constraining disabled children from school attendance can be diverse due to their varied functional difficulties.

On the other hand, disabled children can also be constrained in learning numeracy skills due to diversified challenges in the learning process, even if they are equally enrolled in school. Some literature indicates that children with developmental delay in motor coordination, severe delay in motor skills, and visual-motor integration skills have challenges in learning math [28, 29]. "Embodied cognition" theory argues that the mathematical cognitive process is grounded in the simulations of sensorimotor processes through the interaction of the body with the world [15].

Earlier studies have not supported that children's seeing or hearing abilities are prerequisites for developing essential numeracy competencies. Zarfaty et al. (2004) conclude that deaf children in their early years do not have a problem with representing numbers and are particularly good at representing numbers when sets are presented as spatial arrays [30]. Morgan et al. (2011) also find that children with speech-language impairments do not lag behind non-disabled children in their math skills growth [31]. Crollen et al. (2021) have reported that blind children might even outperform their non-blind peers in numeracy abilities [32]. However, Zhang et al. (2019) demonstrate that children with seeing or auditory perception challenges struggle to learn numeracy skills related to visual Arabic or verbal modules [33].

Numerous studies have also presented evidence that children's development in various abilities, such as information processing, cognitive abilities, and attentive behaviours, is critical for their learning process [22, 34]. Children with intellectual disabilities are often characterised by cognitive, behavioural, and emotional difficulties [35], which can constrain children's ability to learn numeracy skills [34].

Finally, a lack of teaching materials (such as braille or eyeglasses, hearing aids equipment, walking equipment, and sign language) and proper pedagogical interventions for children with disabilities may also constrain their skill learning. Other potential challenges in school can be stigma and negative perceptions, attributions, and expectations of their teachers [36]. Children with multiple disabilities have higher risks related to all the challenges discussed earlier than children with single disabilities. The question is whether or to what extent disabled children's numeracy skills are influenced by factors other than their school attendance and how these factors correlate with their disability status and disability types.

Another concern in investigating children's learning outcomes is school quality [37, 38]. Heyneman & Loxley (1983) studied 29 high- and low-income countries and concluded that in low-income countries, the effect of school quality on primary school children's academic achievement was more prominent than the effect of family socioeconomic status [39]. Bakhshi, Babulal, and Trani (2018) report that when the overall school learning is poor in a conflict setting, there is no difference in learning performance as everybody may lag in poor-quality schools [6]. So far, little evidence in African countries has indicated whether children with disabilities might benefit more or less from high school quality and whether the gap between children with and without disabilities will expand or stabilise when school quality improves. We suggest testing this by comparing the disability effects on children's numeracy skills performance across countries with low- and high-numeracy skills.

Our framework is presented in Fig 1. This paper will estimate the heterogeneous disability effect on the return to schooling regarding numeracy skills with IV models. S1 Table presents the sample size for the split samples. We will estimate the disability effect in the low- and high-numeracy skills country groups for the three disability types, respectively. Ideally, we would have included all five disability types. However, the sample is too small for vision and hearing disabilities in the split sample of sub-groups.

We aim to test the following hypotheses:

H1. *There is a considerable variation in average school performance, measured by the average numeracy skills of children across African countries.*

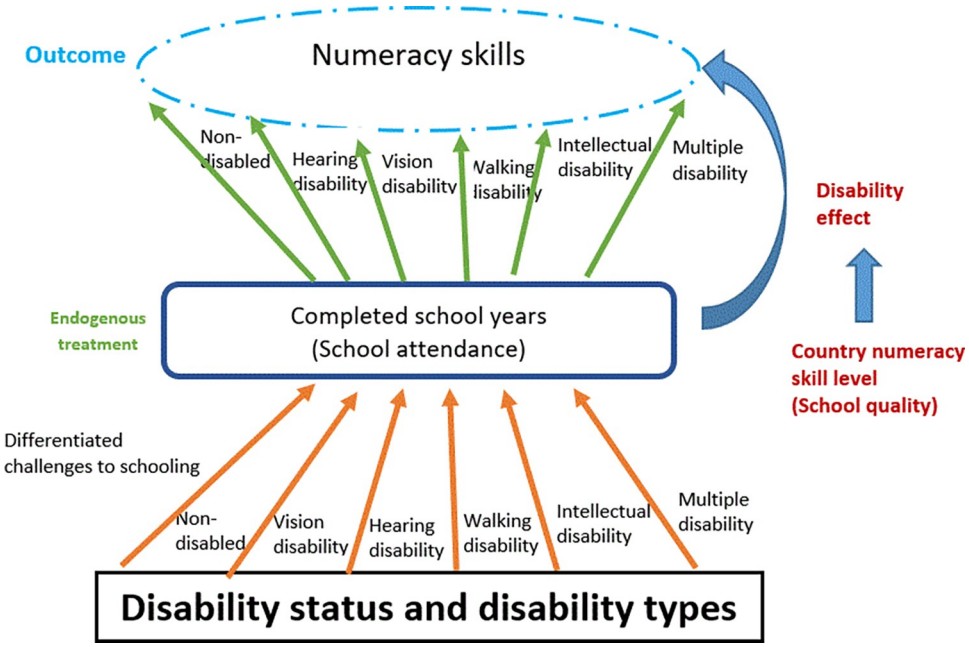

**Fig 1. Framework on numeracy skills performance for children with and without disability.**

H2. *Children with disabilities perform significantly worse than their non-disabled peers of the same age in learning numeracy skills.*

H3. *There are heterogeneous disability effects in numeracy skills return to schooling for children with different disabilities.* More specifically, we hypothesise that:

H3a. *Children with vision and hearing disabilities perform well in numeracy skills return to schooling compared to non-disabled children.* This hypothesis is based on earlier evidence [30, 32], suggesting that vision and hearing abilities might not be crucial in developing numeracy skills. Although learning numeracy skills related to visual or verbal modules might be relevant [33], the numeracy tests involved in this survey are pretty basic.

H3b. *Children with physical disabilities have a lower return to schooling in numeracy skills learning than non-disabled children.* This hypothesis is based on the embodied cognition theory [40] that motor skills can constrain children's numeracy skills learning.

H3c. *Children with intellectual disabilities have a lower return to schooling in numeracy skills learning than non-disabled children.* This hypothesis follows various research findings [22, 34, 35] that children's cognitive and behavioural abilities development is crucial for their numeracy learning.

H3d. *Children with multiple disabilities have the lowest numeracy skills return to schooling among all disability types.* Children with multiple disabilities are exposed to higher challenges [27] because they have fewer opportunities of substituting across senses and functions in their learning processes.

H4a. *The gap in numeracy skills between non-disabled and disabled children is larger in high-numeracy skills countries.* It is based on the assumption that children with disabilities are less capable of benefiting from the better quality of the school system than non-disabled children. Disabled children likely need to give extra effort to the senses and functions that work well to compensate for their disability constraints. More resources and teaching skills are needed to cater for the unique needs of disabled children.

H4b. *The within-group average differences in the numeracy skills between non-disabled and disabled children are smaller than the between-group average differences between the low- and high-numeracy skills country groups.* This is based on the assumption that despite the functional challenges among disabled children, schooling with good quality may greatly contribute to the school performance of children both with and without disabilities.

## 3. Data, methods, and estimation strategy

The MICS surveys aim to provide internationally comparable data about the education status of children and women. Our sample is a national representative sample from eight African countries (DR Congo, The Gambia, Ghana, Lesotho, Sierra Leone, Togo, Tunisia, and Zimbabwe) that conducted the sixth round of the MICS surveys in 2017–2019, conducted by United Nations International Children's Emergency Fund (UNICEF). The survey includes a standard learning assessment test for children aged 7–14 [41].

The MICS survey adopts Washington Group Child Functioning Module (WG-CFM), which aims to capture the most common functional problems related to child development for children aged 6–17 [42, 43]. WG-CFM include 26 questions (related to 13 functional domains) with four severity scales. This paper formulates vision disability as severe difficulty in vision even with glasses or contact lenses; hearing disability as extreme difficulty in hearing even with

a hearing aid; physical disability as severe difficulty in self-care or walking 500 metres on level ground without equipment; intellectual disability as severe difficulties in communication, learning, remembering or concentrating on activities that the child enjoys doing; and multiple disabilities as more than one co-occurring severe functional difficulties as prescribed earlier. Here, severe functional difficulty refers to a lot of functional difficulties or no function at all. The study only uses eight (categorised as four disability types) out of the 13 functional domains captured by WG-CFM. The remaining five functional domains (accepting change, controlling behaviour, making friends, anxiety, and depression) are not included since their prevalence rates vary greatly across the eight African countries, possibly indicating a challenge in interpreting these functional challenges in the local context.

The sample includes currently-in-school, dropout, and never-in-school children. Table 1 shows the total sample size is 32,306, including 30,013 non-disabled children as the counterfactual and 2,293 disabled treatment sample. School enrolment is lower among disabled children (87.8%) than non-disabled children (91.0%) and lowest among multiple disabled children (70.5%). The response rates to the numeracy test among different groups of Ever-In-School children are generally quite high (about 95% or higher) but much lower among the Never-In-School disabled sample (76.1%).

We frame our econometric analysis as a natural experiment, which assumes that the subjects are exposed to a random disability treatment determined by nature or factors outside the control of the subjects or researchers [44]. Disability can be considered an exogenous treatment variable since it is most likely not determined by the characteristics of the population or geographic, economic or social aspects. Despite the potential correlation between poverty and childhood disability declared by some studies [45], the nature of this connection has been complicated. Two mechanisms coexist: children in poor households can be exposed to a higher risk of being disabled, while families with disabled children might experience social deprivation due to the high costs related to their healthcare needs [46]. Moreover, some studies suggest that the gaps in socioeconomic characteristics between people with and without disabilities might be limited and are not statistically significant in a poor environment [47, 48]. To critically assess our natural experiment assumption, we further regress each disability type on a set of individual, family, wealth, and geographical variables (S2 Table). It supports our natural experiment assumption if we find no or very low correlation between these. S2 Table shows that the natural experiment assumption is supported.

**Table 1. Sample size by school status and disability status.**

|  | A | B | C | D | E | F | G | H | I |
|---|---|---|---|---|---|---|---|---|---|
|  | Total sample | Ever-In-School Children (EISC) | % EISC (B/A) | EISC took numeracy test | % EISC took numeracy test (D/B) | Never-In-School Children (NISC) | % NISC (F/A) | NISC took numeracy test | % NISC took numeracy test (H/F) |
| Non-disabled | 30,013 | 27,305 | 91.0 | 26,556 | 97.3 | 2,708 | 9.0 | 2,563 | 94.6 |
| Disabled | 2,293 | 2,013 | 87.8 | 1,922 | 95.5 | 280 | 12.2 | 213 | 76.1 |
| Vision disability | 168 | 163 | 97.0 | 154 | 94.5 | 5 | 3.0 | 5 | 100.0 |
| Hearing disability | 96 | 87 | 90.6 | 81 | 93.1 | 9 | 9.4 | 6 | 66.7 |
| Physical disability | 422 | 357 | 84.6 | 347 | 97.2 | 65 | 15.4 | 54 | 83.1 |
| Intellectual disability | 1,366 | 1,236 | 90.5 | 1,194 | 96.6 | 130 | 9.5 | 114 | 87.7 |
| Multiple disabilities | 241 | 170 | 70.5 | 146 | 85.9 | 71 | 29.5 | 34 | 47.9 |
| Total | 32,306 | 29,318 | 90.8 | 28,478 | 97.1 | 2,988 | 9.2 | 2,776 | 92.9 |

Our outcome variable in this study is children's performance in a numeracy skills test, which is measured as the mean numeracy test score based on four sets of altogether 21 numeracy test questions on symbols reading, quantity comparison, addition and, logical sequence. Our exogenous "treatment" sample consists of children classified in one of the five severe disability types (seeing, hearing, physical, intellectual, and multiple disabilities). The counterfactual sample includes those who did not report severe or moderate disabilities. The disparities in the numeracy test between treatment and control children are assumed to be the treatment impacts or causal disability effects.

The majority of our sample consists of non-disabled children; therefore, we can test hypothesis H1 by assessing the variation in numeracy skills within and across countries. The non-disabled children's performance also serves as a good benchmark to evaluate the numeracy performance of disabled children that are much fewer in number. The fact that we found sizeable across-country variation in numeracy scores among non-disabled children caused us to split our sample into low- and high-numeracy skill countries. We assess the relative performance of non-disabled children versus disabled children within these country groups. This split also serves as a proxy measurement of school quality across the two groups to evaluate the role of school system quality on numeracy skills for disabled children and the gaps between disabled and non-disabled children.

Most studies on disabled children's education apply bivariate or multivariate logistic or probit models to evaluate their access to education (such as school enrolment, school completion, dropout, highest grade achieved) for children with and without disability [17–19, 26]. Some studies simply use univariate analysis while including the disability status [15, 21]. Some studies dichotomise the indicators (able to read or write) for school performance and use logistic or probit models [6, 25]. Takeda and Lamichhane (2018) use an OLS model to estimate school performance as a continuous score [24]. These studies assess the correlation between children's disability status and their school performance without considering the cause-effect of disability on children's schooling. A few studies use household fixed-effect models to estimate the disability effect by controlling for unobserved and observed household characteristics [17, 18]. However, such kind of studies require a sample of children both with and without disabilities from the same household, which may not always be available.

Children's numeracy skills are primarily learnt through school attendance. Disabled children may fall behind other children in numeracy skills for two reasons. First, they may fall behind because they cannot attend school and complete fewer school years. Second, their disability may limit their numeracy skill learning ability while in school. Children's educational attainment (completed school years) can be considered as both the outcome of disability and, at the same time, an endogenous treatment on skill learning. Therefore, to estimate the disability effect of numeracy skills, we suggest using the instrumental variable (IV) method to control for the potential bias associated with endogenous completed school years of disabled versus non-disabled children.

In the first set of regressions (Eq (1) below), we test hypothesis H2, which states that children with disabilities perform significantly worse than their non-disabled peers of the same age in learning numeracy skills. We first test a reduced-form model which ignores the causal mechanisms with a parsimonious specification. The first model includes only age and the treatment variable $D_{ij}$, indicating children as non-disabled or with disability type $j$. We then run additional models, first including the country dummies and then gender. Without considering endogenous treatment and possible interaction effects, the first set of regressions allows us to assess the variation in numeracy skills by age and disability types.

$$Numeracy_i = \beta_0 + \beta_{1j}D_{ij} + \beta_2 Age_i + \beta_3 Gender_i + \beta_{4k}Country_{ik} + u_{ijk} \qquad (1)$$

Here, the subscript $i$ represents each individual child, $j$ represents a type of disabilities (including children without disability, children with vision, hearing, physical, intellectual, and multiple disabilities), $k$ represents countries, and $u_{ijk}$ is the error term. In the models, $\beta_0$ estimates the average score rates of numeracy tests for the 7-year-old non-disabled control children in DRCongo (the country used as the base). $\beta_{1j}$ estimates the marginal disability treatment effects of disability type j on children's performance of numeracy skills.

In the second set of regressions, we want to test hypothesis H3, which suggests heterogeneous disability effects in learning numeracy skills for children with different disabilities. The type of disability may affect each step in the causal mechanisms in different ways; therefore, we run the IV models on the split samples by various disability statuses:

$$\text{Outcome equation}: \ Numeracy_{ij} = \Upsilon_{1j} * CSY_{ij} \tag{2}$$

$$\text{Selection equation}: \ CSY_{ij} = \pi_{0j} + \pi_{1j}\ln(Age_{ij}) + \pi_{2j}Gender_{ij} + \varepsilon_{ij} \tag{3}$$

Here, $\Upsilon_{1j}$ estimates the average numeracy skills return to each completed school year among the children with disability type $j$. This is the parameter of interest. We want to test whether the return to education per school year in numeracy skills is homogeneous or depends on disability types. In the first stage of regressions, $\pi_{1j}$ and $\pi_{2j}$ capture the effect of age and gender on the number of school years completed by children with disability type j. ln(age) is included since it performs best in satisfying the Sargan overidentification test. The constant term $\pi_{0j}$ is included in the first stage but not the second one since we assume that children learn numeracy skills mainly from school and therefore have no numeracy skills when they start school. We apply the *ivregress 2SLS* estimator in Stata 15.

In the IV model, to satisfy the theoretical validity of our identification strategy, we use age and gender as instruments, as these variables affect completed school years. They do not directly affect numeracy skills learning (exclusion restriction). For children's age and gender to be strong instruments, they must be strongly correlated with the completed school years. For these instruments to be statistically valid, they must be uncorrelated with the error term in the numeracy skills (outcome) model. These properties are also statistically testable in the overidentified case. We present standard IV instrument tests of endogeneity (Robust Wu-Hausman test), the strength of the instruments (first stage F test), and the overidentification (Sargan IV validity test). We also present results from Ordinary Least Square (OLS) regressions if the IV tests are not satisfied.

In the third set of regressions, we want to test hypotheses H4a and H4b, which evaluate the role of school system quality on the numeracy skills of disabled children and the gaps between disabled and non-disabled children. We run all IV split-sample models in the low- and high-numeracy skills country groups for the non-disabled children and children with physical, intellectual, and multiple disabilities, respectively.

## 4. Results

### 4.1 Descriptive analysis

The descriptive statistics of outcome and control variables are presented in S3 Table. We calculate children's overall numeracy test scores as the mean value of 21 numeracy questions (0 = wrong, 1 = correct). We show the mean test scores by children's age (left figure) and by completed school years (right figure) in Fig 2. The figure draws vertical box plots, which show the median, 25th and 75th percentile (upper and lower hinge) and lower and upper adjacent values of the mean test scores in each group. The outside values are plotted as dots. The figure

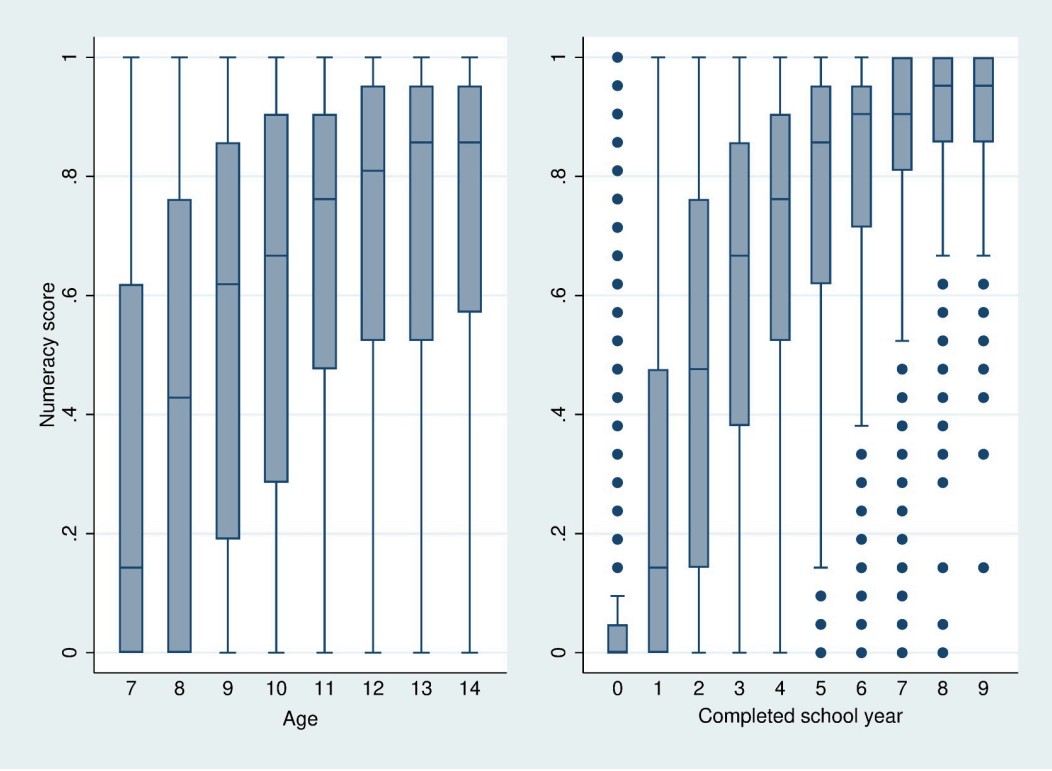

**Fig 2. Numeracy test scores by children's age or by completed school years (median, 25th and 75th percentile).**

suggests that children perform better in numeracy skills when they grow older. The disparity in numeracy skills performance by completed school years is higher than the age disparity. It is in line with the earlier assumption that age does not directly affect numeracy skills and only involves exposure to schooling.

Table 2 shows the mean numeracy score by countries for non-disabled and disabled children, respectively. The overall mean numeracy score for the non-disabled is 0.57, which is relatively low in DRCongo (0.35), Sierra Leone (0.41) and The Gambia (0.50). In the remaining five countries (Ghana, Lesotho, Togo, Tunisia, and Zimbabwe), the mean numeracy score is between 0.63 and 0.88. The average numeracy skills in these countries are about double those

**Table 2. Mean numeracy score by countries.**

| | Non-disabled | | Disabled | | Sample size | | |
|---|---|---|---|---|---|---|---|
| | Mean | Std. err. | Mean | Std. err. | Non-disabled | Disabled | Total |
| DRCongo | 0.35 | 0.004 | 0.25 | 0.014 | 6268 | 395 | 6,663 |
| The Gambia | 0.50 | 0.007 | 0.37 | 0.033 | 3104 | 128 | 3,232 |
| Ghana | 0.70 | 0.005 | 0.59 | 0.015 | 4372 | 542 | 4,914 |
| Lesotho | 0.68 | 0.006 | 0.57 | 0.029 | 2567 | 141 | 2,708 |
| Sierra Leone | 0.41 | 0.005 | 0.36 | 0.019 | 4761 | 324 | 5,085 |
| Togo | 0.63 | 0.007 | 0.57 | 0.023 | 2252 | 202 | 2,454 |
| Tunisia | 0.88 | 0.004 | 0.73 | 0.025 | 2135 | 168 | 2,303 |
| Zimbabwe | 0.75 | 0.005 | 0.63 | 0.025 | 3660 | 235 | 3,895 |
| **Total** | **0.57** | **0.002** | **0.49** | **0.008** | **29,119** | **2,135** | **31,254** |

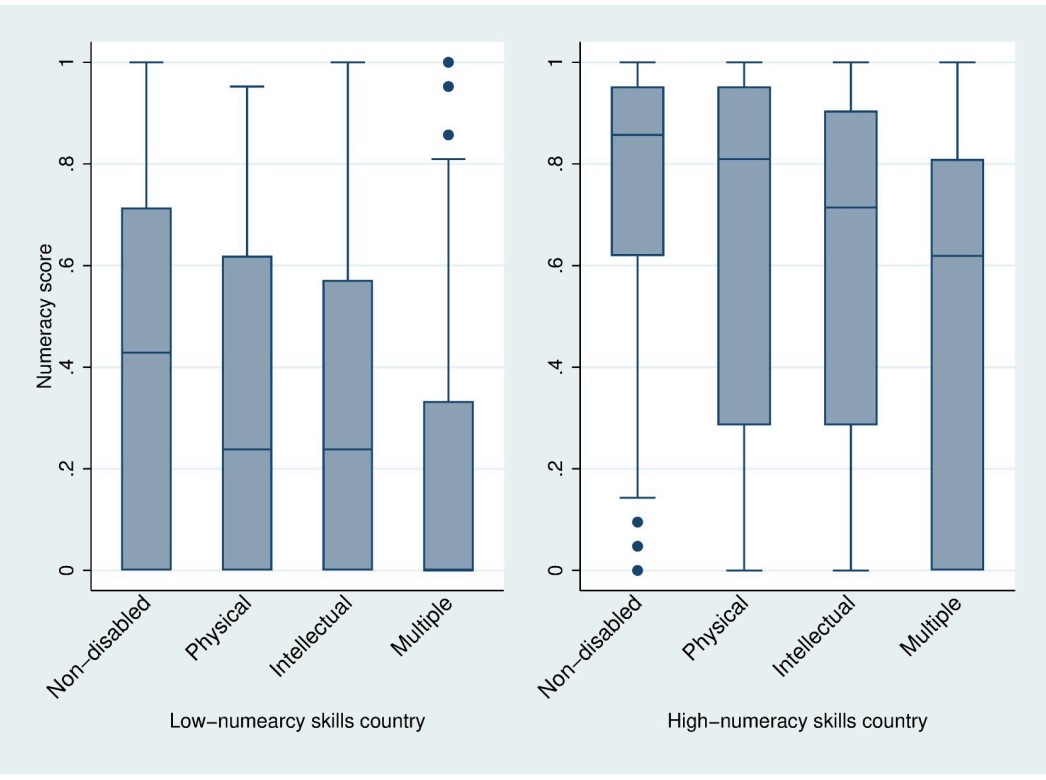

**Fig 3. Numeracy test scores in low- and high-numeracy skills countries (median, 25th and 75th percentile).**

in DRCongo. The mean numeracy scores in DRCongo and Sierra Leone are significantly lower than all the five countries with higher scores. Hypothesis H1 on the large variation in average numeracy skills performance among children across African countries is supported. We suggest dividing our sample into two groups: the low-numeracy countries group (DRCongo, Sierra Leone, and The Gambia) and the high-numeracy country group (Ghana, Lesotho, Togo, Tunisia, and Zimbabwe).

Table 2 shows that non-disabled children answered 57% of the questions correctly, and the disabled sample answered 49% correctly. The descriptive statistics in S3 Table demonstrate that children with hearing and vision disabilities answered more questions correctly than non-disabled children. In contrast, the correct response rates for children with other disabilities are much lower. We present the test score distributions (median, p25, and p75) for the low-numeracy countries (left figure) versus the high-numeracy countries (right figure) by disability types as vertical box plots in Fig 3. The mean test scores with 95% confidence intervals by disability type are presented in Fig 4. With the split sample, the sample size is too small for reliable statistical analysis for children with vision and hearing disabilities, as shown in S1 Table. Therefore, we restrict our split sample analysis to children with physical, intellectual, and multiple disabilities.

Figs 3 and 4 indicate the significant disparities in numeracy tests not only between the two groups of countries but also between children with and without disabilities. Disabled children lag in numeracy skills performance in both groups of countries. However, descriptive data suggest that disabled children benefit from improved school quality since disabled children in high-numeracy skills countries perform even better than non-disabled children in low-numeracy skills countries. The question is whether disabled children gain as much as non-disabled children in learning numeracy skills when the learning environment has improved.

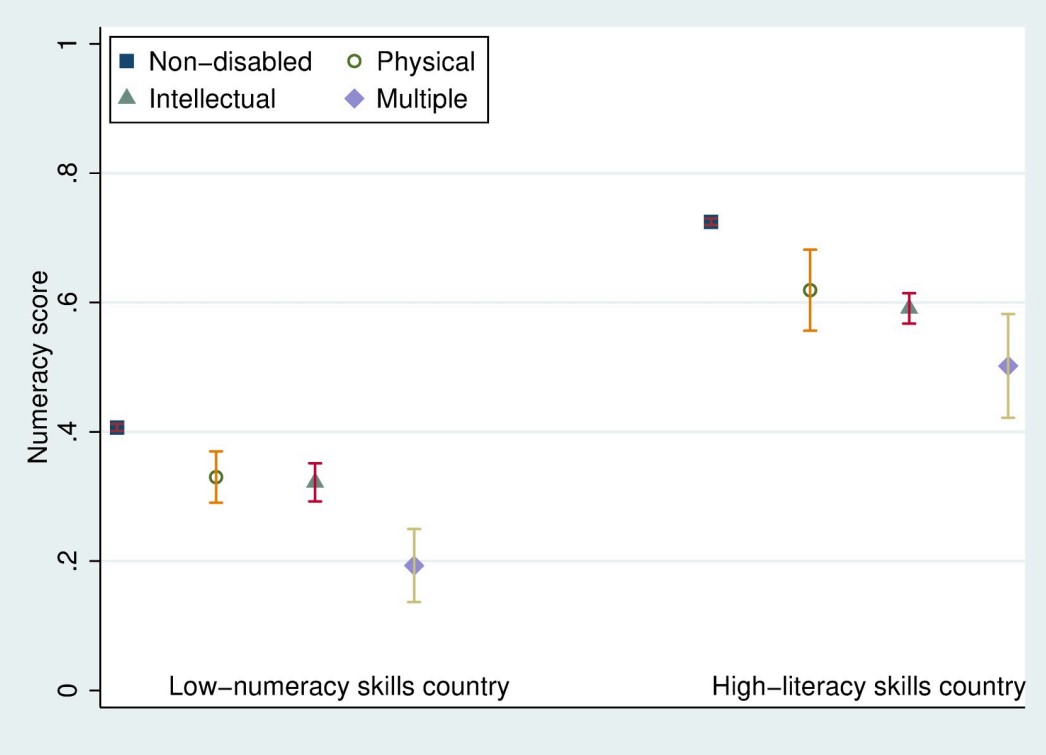

**Fig 4. Mean numeracy test scores with 95% confidence intervals for the means in low- and high-numeracy skills countries.**

### 4.2 Disability effect with age control

The first set of regressions aims to test hypothesis H2, which states that children with disabilities perform significantly worse than their non-disabled peers of the same age in learning numeracy skills. Without considering the causal mechanisms, we start with a parsimonious specification, including age, country and gender dummy variables stepwise as control variables. The regression results are presented in Table 3.

The constant term in Model 1 suggests that the estimated average score is 0.31 for 7-year-old control children. Children's numeracy skills improve with age, probably related to their access to schooling. Model 2 shows effective numeracy skills variation across countries. To evaluate the numeracy skills gap over countries, we run separate regressions with age dummies for non-disabled children in each country (S4 Table). DRCongo is the country with the lowest numeracy skills, where the average numeracy score is only 0.106 for 7-year-old children, while Tunisia has the highest average numeracy score of 0.77 for 7-year-old children. The country dummy variable parameters and their significance levels illustrate large variations in school quality across countries in their performance in enhancing children's average numeracy skills. Finally, gender is not significantly correlated with children's numeracy skills performance. It indicates that girls are not discriminated against in the school systems in a way that affects their basic numeracy skills.

The coefficients on the disability status in model 1 in Table 3 show a significant and negative disability effect on children's numeracy skills for children with physical, intellectual, and multiple disabilities. However, the estimated disability effect for children with physical disabilities turns insignificant after controlling for the macro country dummy (models 2 and 3). In

**Table 3. Regression results for disability effects on the mean numeracy test score.**

| | Model 1 | Model 2 | Model 3 |
|---|---|---|---|
| **Disability status** | | | |
| Vision disability | 0.121*** | 0.028 | 0.029 |
| | (0.024) | (0.021) | (0.021) |
| Hearing disability | -0.002 | -0.049 | -0.049 |
| | (0.036) | (0.031) | (0.031) |
| Physical disability | -0.068*** | -0.019 | -0.019 |
| | (0.020) | (0.017) | (0.017) |
| Intellectual disability | -0.072*** | -0.109*** | -0.109*** |
| | (0.010) | (0.009) | (0.009) |
| Multiple disabilities | -0.213*** | -0.205*** | -0.205*** |
| | (0.026) | (0.024) | (0.024) |
| **Age** | | | |
| 8 | 0.128*** | 0.127*** | 0.127*** |
| | (0.007) | (0.006) | (0.006) |
| 9 | 0.242*** | 0.231*** | 0.231*** |
| | (0.007) | (0.007) | (0.007) |
| 10 | 0.284*** | 0.277*** | 0.277*** |
| | (0.007) | (0.006) | (0.006) |
| 11 | 0.355*** | 0.337*** | 0.337*** |
| | (0.008) | (0.007) | (0.007) |
| 12 | 0.382*** | 0.371*** | 0.371*** |
| | (0.007) | (0.007) | (0.007) |
| 13 | 0.420*** | 0.398*** | 0.398*** |
| | (0.007) | (0.006) | (0.006) |
| 14 | 0.439*** | 0.415*** | 0.415*** |
| | (0.007) | (0.006) | (0.006) |
| **Country (base category: DRCongo)** | | | |
| The Gambia | | 0.147*** | 0.147*** |
| | | (0.012) | (0.012) |
| Ghana | | 0.327*** | 0.327*** |
| | | (0.010) | (0.010) |
| Lesotho | | 0.298*** | 0.298*** |
| | | (0.009) | (0.009) |
| Sierra Leone | | 0.066*** | 0.066*** |
| | | (0.010) | (0.010) |
| Togo | | 0.274*** | 0.274*** |
| | | (0.011) | (0.011) |
| Tunisia | | 0.501*** | 0.501*** |
| | | (0.008) | (0.008) |
| Zimbabwe | | 0.396*** | 0.396*** |
| | | (0.008) | (0.008) |
| **Gender (1 = girl, 0 = boy)** | | | 0.003 |
| | | | (0.003) |
| **Constant** | 0.305*** | 0.106*** | 0.105*** |
| | (0.005) | (0.007) | (0.007) |
| **Sample size** | 31254 | 31254 | 31254 |

(*Continued*)

**Table 3.** (Continued)

|        | Model 1 | Model 2 | Model 3 |
|--------|---------|---------|---------|
| **R2** | 0.171   | 0.373   | 0.373   |

Significance levels: * p<0.05; ** p<0.01; *** p<0.001

contrast, it becomes larger for children with intellectual disabilities after controlling for the country dummy. The country effect might be important for evaluating the disability effect for children with physical and intellectual disabilities. The first set of regressions supports hypothesis H2 that children with physical, intellectual, and multiple disabilities perform significantly worse than their non-disabled peers of the same age in learning numeracy skills. However, hypothesis H2 is not supported for children with vision and hearing disabilities.

### 4.3 IV models with endogenous completed school years

We will now more closely study the causal mechanisms for the links between the exogenous disability (treatment) variables and the outcome. The disability effect on numeracy skills may come from reduced school participation or a lower ability to acquire numeracy skills while in school. To analyse this, we run IV models with completed school years as the endogenous exposure to schooling on the split samples for each disability type.

We run IV models with age and gender as instruments. To test the strength of the two instruments and assess the endogeneity of completed school years in the model, we first run a set of regressions, presented in S5 Table. All the models in S5 Table suggest that age and gender are significantly associated with the completed school years. Moreover, the disability effects on completed school years vary a lot across disability types, which suggests potentially high endogeneity of the completed school years. Furthermore, the regressions in section 4.2 suggest that gender does not directly affect children's numeracy skills.

The regression results and IV tests are shown in Table 4. The OLS model results are presented for the non-disabled when the IV tests are invalid. For the models that satisfy the tests, we find the following results. The first-stage regression indicates that children with vision or hearing disabilities do not lag in completed school years compared to non-disabled children. However, children with physical, intellectual, or multiple disabilities have completed significantly fewer school years than non-disabled children per year of age.

The return to each completed school year in numeracy skills score is estimated at 0.146 units for non-disabled children in the IV model and 0.142 in the OLS model, noting that the overidentification test failed for this IV model. For the other IV models, the statistical validity could not be rejected. For children with vision, hearing, physical, and intellectual disabilities, there is no significant disability effect on numeracy skills returns to completed school years. Hypothesis H3a, which states that children with vision and hearing disabilities perform well in numeracy skills return to schooling compared to non-disabled children, cannot be rejected. However, H3b and H3c, which state that children with physical or intellectual disabilities have a lower return to schooling in numeracy skills than non-disabled children, are not supported.

The estimated return to each completed school year is 0.142 (CI:0.140–0.144) for non-disabled children and 0.121 (CI:0.105–0.137) for children with multiple disabilities. Significant disability effects of 0.121–0.142 = -0.021 score points for each completed school year are reported for children with multiple disabilities. Hypothesis H3d that children with multiple disabilities have the lowest return to schooling regarding numeracy skills cannot be rejected.

**Table 4. Regressions on the mean numeracy score by disability types.**

| | OLS for non-disabled | IV (separate model for each disability type) | | | | | |
|---|---|---|---|---|---|---|---|
| | | Non-disabled | Vision disabled | Hearing disabled | Physical disabled | Intellectual disabled | Multiple disabled |
| Completed school years (base category: 1) | 0.142*** | 0.146*** | 0.147*** | 0.143*** | 0.151*** | 0.145*** | 0.121*** |
| | (0.000) | (0.001) | (0.005) | (0.009) | (0.006) | (0.002) | (0.008) |
| Sample size | 29119 | 29119 | 159 | 87 | 401 | 1308 | 180 |
| First stage regressions (Dep: Completed school year) | | | | | | | |
| Ln(age) | | 7.599*** | 8.440*** | 7.970*** | 6.523*** | 6.731*** | 5.637*** |
| | | (0.044) | (0.376) | (0.641) | (0.379) | (0.209) | (0.637) |
| Gender (1 = girl, 0 = boy) | | 0.021 | 0.108 | 0.340 | 0.106 | -0.025 | -0.020 |
| | | (0.019) | (0.193) | (0.336) | (0.136) | (0.090) | (0.287) |
| Constant | | -13.954*** | -15.371*** | -15.064*** | -11.902*** | -12.268*** | -10.203*** |
| | | (0.092) | (0.838) | (1.444) | (0.789) | (0.450) | (1.374) |
| IV test | | | | | | | |
| Robust Wu-Hausman test (p value) | | 0.000 | 0.000 | 0.000 | 0.000 | 0.000 | 0.060 |
| Sargan IV validity test (p-value) | | 0.000 | 0.960 | 0.989 | 0.560 | 0.349 | 0.126 |
| Strength (First stage F test) | | 21324.1 | 381.4 | 145.54 | 302.73 | 1497.77 | 149.46 |

Instrumented: Completed school year. Instruments: ln(age) and gender dummy.

Significance levels: * $p < 0.05$; ** $p < 0.01$; *** $p < 0.001$, based on the standard errors which allow for intragroup correlation

## 4.4 IV models for low- and high-numeracy skills countries

The results in Table 3 show that there might be a country effect when evaluating the overall disability effect for children with physical and intellectual disabilities. This might indicate heterogeneous disability effects across the eight African countries. To further explore the disability effects for different disability types, we run IV regressions after dividing the sample into low- and high-numeracy skills country groups as defined in section 4.1. The sample sizes of the split samples by country numeracy skills level and disability status only allow for the analyses of three disability types (physical, intellectual, and multiple disabled). The regressions are run on the split samples of the non-disabled and disabled for each of the three specific disability statuses in the countries with low and high numeracy skills, respectively. The results are presented in Table 5.

We then graph the regression coefficients with 95 per cent confidence intervals to present the first stage estimated completed school year by age (Fig 5) and the second stage estimated numeracy skills return to completed school years (Fig 6) over different disability types in low- and high-numeracy skills country groups. Fig 5 indicates that in both groups of countries, the mean estimated completed school years by age for intellectually disabled children and multiple disabled children are significantly lower than those for non-disabled children. Children with physical disabilities have also completed fewer school years than non-disabled children, but the differences are not significant. The gap in completed school years in the low-numeracy skills country group is higher than in the high-numeracy skills group.

Fig 6 suggests that the mean estimated numeracy skills return to each completed school year in low-numeracy skills countries is 0.132 (CI: 0.130–0.134) score points for non-disabled children. In contrast, it is estimated to be 0.152 (CI: 0.150–0.154) score points in the high-numeracy skills country group. Children with physical or intellectual disabilities are not significantly different from non-disabled children in numeracy skills return to schooling. In contrast, the mean estimated numeracy returns are 0.107 (CI: 0.082–0.132) and 0.129 (CI: 0.111–

**Table 5. Regressions on the mean numeracy score in low- and high-numeracy skills country group.**

| | Low-numeracy skills group | | | | | High-numeracy skills group | | | | |
| --- | --- | --- | --- | --- | --- | --- | --- | --- | --- | --- |
| | OLS for intellectual disabled | IV (separate model for each disability type) | | | | OLS for non-disabled | IV (separate model for each disability type) | | | |
| | | Non-disabled | Physical disabled | Intellectual disabled | Multiple disabled | | Non-disabled | Physical disabled | Intellectual disabled | Multiple disabled |
| Completed school years (base category: 1) | 0.128*** | 0.132*** | 0.141*** | 0.138*** | 0.107*** | 0.152*** | 0.155*** | 0.166*** | 0.148*** | 0.129*** |
| | (0.005) | (0.001) | (0.009) | (0.005) | (0.013) | (0.001) | (0.001) | (0.009) | (0.003) | (0.009) |
| Sample size | 435 | 14133 | 268 | 435 | 93 | 14986 | 14986 | 133 | 873 | 87 |
| First stage regressions (Dep: Completed school year) | | | | | | | | | | |
| Ln(age) | | 6.580*** | 5.782*** | 4.738*** | 3.664*** | | 8.330*** | 7.731*** | 7.376*** | 6.977*** |
| | | (0.075) | (0.479) | (0.364) | (0.983) | | (0.046) | (0.542) | (0.222) | (0.778) |
| Gender (1 = girl, 0 = boy) | | -0.042 | 0.096 | -0.014 | -0.030 | | 0.135*** | 0.349 | 0.098 | -0.154 |
| | | (0.029) | (0.163) | (0.150) | (0.358) | | (0.021) | (0.209) | (0.098) | (0.332) |
| Constant | | -12.183*** | -10.581*** | -8.523*** | -6.530*** | | -15.142*** | -14.054*** | -13.434*** | -12.378*** |
| | | (0.156) | (0.988) | (0.762) | (2.159) | | (0.097) | (1.119) | (0.487) | (1.647) |
| IV test | | | | | | | | | | |
| Robust Wu-Hausman test (p value) | 0.000 | 0.000 | 0.000 | 0.568 | | 0.000 | 0.000 | 0.000 | 0.004 | |
| Sargan IV validity test (p-value) | 0.656 | 0.655 | 0.004 | 0.694 | | 0.000 | 0.465 | 0.452 | 0.123 | |
| Strength (First stage F test) | 5193.02 | 159.57 | 334.46 | 40.59 | | 24230.81 | 191.2 | 1483.36 | 143.04 | |

Instrumented: Completed school year. Instruments: Ln(age) and gender

Significance levels: * $p<0.05$; ** $p<0.01$; *** $p<0.001$, based on the cluster-robust standard errors

0.147) for children with multiple disabilities in countries with low- and high-numeracy skills. The gap between non-disabled children and children with multiple disabilities in the low numeracy countries -0.025 (= 0.107–0.132) is marginally higher than that -0.023 (= 0.129–0.152) in the high numeracy countries. Furthermore, numeracy skills return to schooling for children with physical 0.166 (CI: 0.148–0.184) or intellectual disabilities 0.148 (CI: 0.142–0.154) in high-numeracy skills countries are significantly higher than that of the non-disabled children 0.132 (CI: 0.130–0.134) in low-numeracy skills countries. It indicates that disabled children benefit as much from higher school quality as non-disabled children do.

Finally, the numeracy skills performance is predicted for a 14-year-old child by disability status in both low- and high-numeracy skills groups in Fig 7. The endogenous school year differences, as well as differences in return to each endogenous school year in both stages, are taken into consideration. The total effects of disability on numeracy skills for 14-year-old children are negative and significant for both intellectual and multiple disabled children in countries with low- and high-numeracy skills. The predicted mean numeracy skill for children with intellectual disability is 0.547 (CI: 0.504–0.590) in low-numeracy skills countries and 0.899 (CI: 0.869–0.930) in high-numeracy skills countries, which is significantly lower than that for non-disabled children of 0.679 (CI: 0.669–0.688) and 1.073 (CI: 1.065–1.081) in low- and high-numeracy skills countries.

The disability effect for children with intellectual disability are -0.13 (= 0.547–0.679) and -0.17 (= 0.899–1.073) in low- and high-numeracy skills countries, and that for children with multiple disabilities are -0.34 and -0.30, respectively. For those with physical disabilities, there is no significant disability effect in low- or high-numeracy skills groups.

The cross-country difference in predicted numeracy skills for a 14-year-old non-disabled child is about 0.4 points between low- and high-numeracy skills country groups, which is

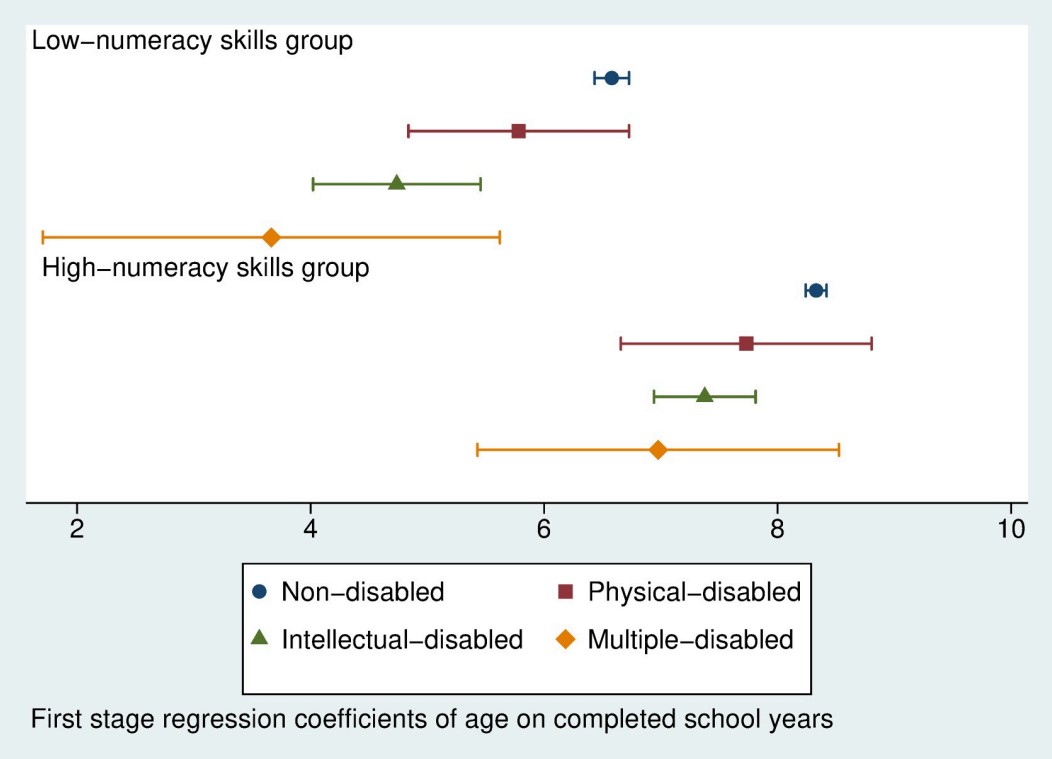

**Fig 5. First-stage regression coefficients of age on completed school years with 95% confidence intervals (IV regression on numeracy skill return to each completed school year, separate IV models for three disability types in low- and high-numeracy skills country groups).**

marginally higher than the estimated numeracy skills gaps across disability types, as discussed above.

Furthermore, 14-year-old children with intellectual disabilities in high-numeracy skills countries show significantly better numeracy skills performance (0.90) than the non-disabled children in the low-numeracy skills group (0.68). The average score of non-disabled children in the low-numeracy skills countries (0.68) is even below the average numeracy score for the most challenged multiple-disabled children in high-numeracy skills countries (0.77).

These findings do not support hypothesis H4a, that children with disabilities are less capable of benefiting from the better quality of the school system than non-disabled children. Disabled children do benefit substantially from improved school quality. The gap between non-disabled and disabled children in numeracy skills is smaller than the variation across countries, which supports hypothesis H4b.

## 5. Discussion

We will now summarise our findings for the key hypotheses and discuss our results related to the relevant literature and earlier studies. The first hypothesis (H1) states a considerable variation in average numeracy skills across the eight African countries we have studied. Our analyses reveal large variations in average numeracy skills across countries based on nationally representative data; thus, we cannot reject this hypothesis. The large sample sizes provide accurate estimates of mean numeracy skill scores by country since they have confidence intervals in the range of 0.01–0.015 around the mean numeracy skills scores, ranging from the lowest

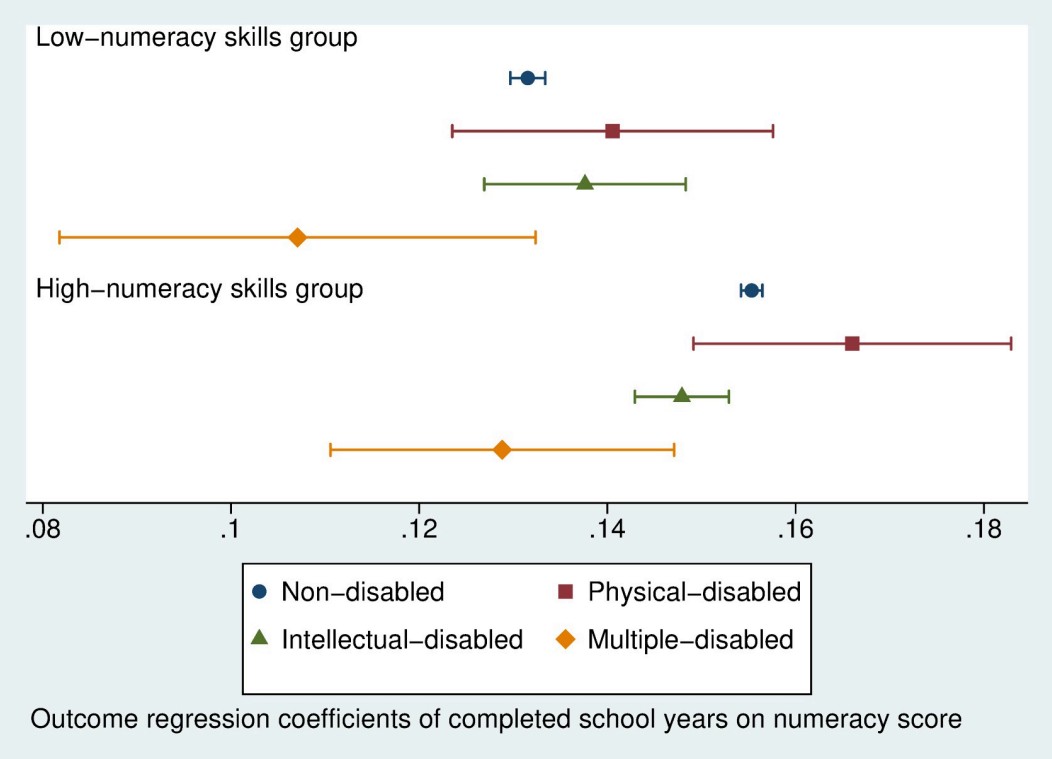

**Fig 6. Outcome regression coefficients of completed school years on numeracy scores with 95% confidence intervals (IV regression on numeracy skill return to each completed school year, separate IV models for three disability types in low- and high-numeracy skills country groups).**

0.35 in DRCongo to the highest 0.88 in Tunisia. It indicates considerable variation in the average quality of school systems across these eight countries regarding their ability to teach children numeracy skills.

Our second hypothesis (H2) that disabled children perform worse than their non-disabled peers in numeracy skills was supported for children with physical, intellectual, and multiple disabilities but not those with vision and hearing disabilities. To our knowledge, almost no similar study has evaluated disabled children's numeracy skills in the African context. The only exception is the study by Bakhshi, Babulal, and Trani (2018) from Sudan [6]. The other few earlier papers in the developing context are mainly from Asia, with the study of Takeda and Lamichhane (2018) from India [24], Malik et al. (2020) and Singal et al. (2020) from Pakistan [25, 26]. Most studies have applied the Washington Group definition of disabilities. Bakhshi, Babulal, and Trani (2018) used a disability screening questionnaire (DSQ-35), and Takeda and Lamichhane (2018) revised the WG module to a large extent. The age range of children included in the learning assessment test also varies. The two studies in Pakistan use the ASER (Annual Status of Education Report) test on reading and math. Takeda and Lamichhane (2018) use reading, math and writing test in the Indian Human Development Survey (IHDS). Bakhshi, Babulal, and Trani (2018) use simple self-reporting assessments. Despite the disparities of these studies, most studies reported a performance gap between disabled and non-disabled children, except the study by Bakhshi, Babulal, and Trani (2018). Our findings provide evidence in the African context, suggesting a gap in numeracy skills between disabled and non-disabled children, which varies across disability types.

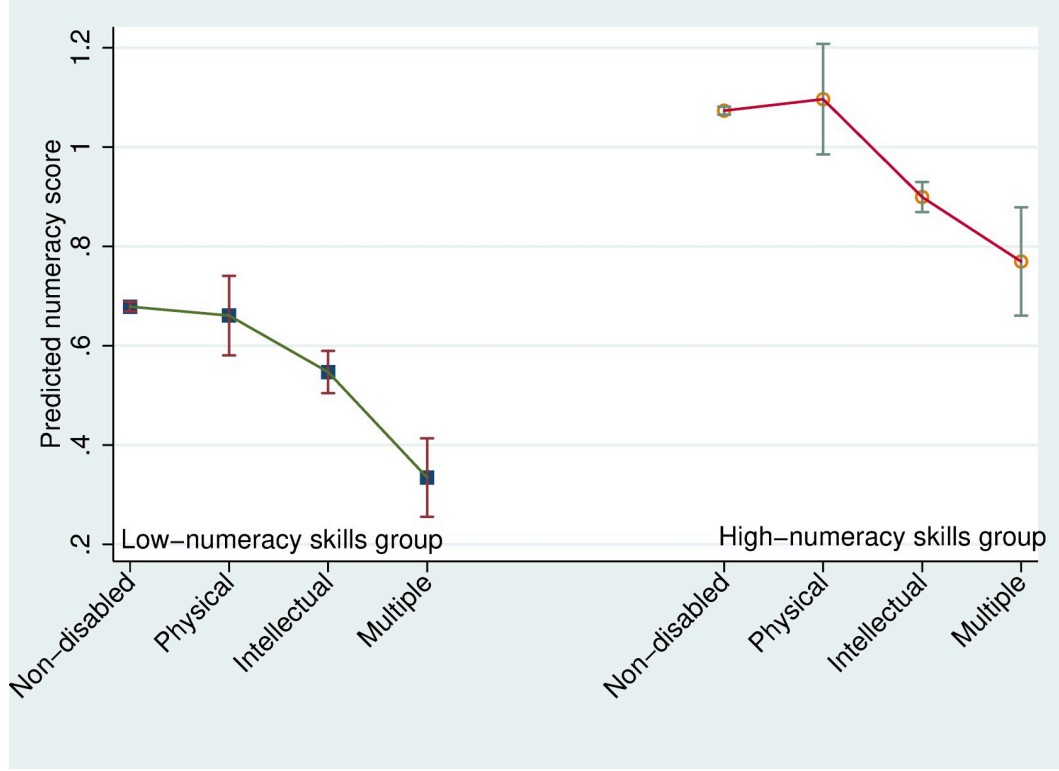

**Fig 7. Predicted numeracy skills performance by disability status for an average 14-year-old child in both low- and high-numeracy skills groups.**

Little empirical evidence has been available for heterogenous disability effects on school performance by disability types in the African context. Our results suggest that children with vision and hearing disabilities do not have lower numeracy skills than non-disabled children, which supports our hypothesis H3a. It is not the case for children with other disabilities. Also, based on the WG definition of disabilities, a study in Pakistan by Singal et al. (2020) is one of the few studies differentiating the disability types, which uses the ASER (Annual Status of Education Report) test [26]. They report that children with moderate or severe sensory disabilities (walking, seeing and hearing) have the lowest level of basic numeracy skills. Singal et al. (2018) transferred the test scale to a very low threshold dichotomy variable and only evaluated whether children could identify one-digit numbers. It might explain the special challenges for children with sensory disabilities compared to children with other disabilities. Our study does not find challenges for children with vision and hearing disabilities, but it does not mean they are not exposed to additional risks in school performance. The numeracy test embedded in the MICS survey might not fully capture the potential risk for those with vision and hearing disabilities to learn more advanced numeracy skills.

Earlier studies on the numeracy skills differences have not specifically differentiated the mechanisms behind possible disability effects. Such effects could simply be caused by the lack of school attendance, or they could be related to disabled children's low returns to schooling in numeracy skills. We separate the two types of disability effects by IV models to control for the endogeneity of completed school years for each disability type. In their study in India, Takeda and Lamichhane (2018) suggest that disabled children are less likely to fall behind in skills once they access education [24]. They made this conclusion because they noticed that when

the interaction between disability status and school status is included in the model, the disability dummy becomes insignificant. However, they did not consider the potential endogeneity of schooling.

Our IV model shows that low numeracy scores among the physical- and intellectually disabled children are mainly attributable to the low school years they manage to complete but are not constrained by their numeracy skills returns to schooling. Hypotheses H3b and H3c state that children with physical or intellectual disabilities have a lower return to schooling in numeracy skills (after controlling for differences in completed school years) compared to non-disabled children. Our results do support the two hypotheses. These findings suggest that school enrolment is especially crucial for children with disabilities to gain equal access to education. On the other hand, children with multiple disabilities have not only completed the least school years but also have the lowest numeracy skill returns per completed school year among children with various disability types, which supports hypothesis H3d.

Finally, hypothesis H4a states that the gap in numeracy skills between non-disabled and disabled children is larger in high-numeracy skills countries. However, our study shows that the overall gap between children with and without disabilities in terms of numeracy skills, considering both effects of endogenous school year differences and differences in school return to each school year, is not significantly different between low- and high-numeracy skills countries. It does not provide evidence of a broader gap in school performance for disabled children when the school quality is improved. Therefore, we reject hypothesis H4a.

Bakhshi, Babulal, and Trani (2018) found in their study in West Darfur of Sudan that when all the children are exposed to low-quality schools in a conflict context, there is no difference in numeracy skills between the disabled and non-disabled children [6]. By controlling the endogeneity of completed school years, we find that in low- and high-numeracy skills countries, most children with disabilities (except children with multiple disabilities) do not lag significantly in gaining numeracy skills if they complete the same schooling as the non-disabled children. Their main challenge is low school enrollment, especially in countries with poor school quality.

The estimated numeracy skills return to schooling among children with physical or intellectual disabilities in high-numeracy skills countries are significantly higher than that of the non-disabled peers in low-numeracy skills countries. The variation in numeracy skills performance is higher across countries than over disability types. We cannot reject hypothesis H4b, that the average numeracy skills of non-disabled children vary more across countries with different school system quality than the gap between non-disabled and disabled children. The variation across countries can be even higher if more countries are included, which suggests the quality of the school system is the key to improving school performance in Africa.

## 6. Conclusion

Based on large-scale nationally representative samples in the eight African countries, we assess the within- and across-country variation in numeracy skills and the gap between children with and without disabilities. The Washington Group Child Functional Module (WG-CFM) and standard numeracy test are applied to all the countries and ensure the credibility of this comparison study. We identify two types of disability effects using IV models to control for the endogeneity of completed school years. These models allow us to divide the numeracy skill differences into the difference in completed school years and the difference in numeracy skill returns per completed school year. Combining these two effects results in systematic variations in the overall numeracy skill performance across disability types.

First, we find a considerable variation in average numeracy skills across the eight African countries (hypothesis H1 is supported). Second, there is systematic variation in numeracy skills across disability types (hypothesis H2 is supported). More specifically, children with vision and hearing disabilities perform as well as non-disabled children, while children with physical, intellectual and multiple disabilities lag behind. Third, the reason why children with different disability types lag behind varies (hypotheses H3a and H3d are supported; hypotheses H3b and H3c are rejected). Those with physical and intellectual disabilities lag because they, conditional on age, have completed fewer school years. Those with multiple disabilities lag both due to fewer completed school years and due to lower numeracy skill returns per school year.

Furthermore, based on average performance, we find that the within-group average differences in numeracy skill returns to school between non-disabled and disabled children are similar between the low- and high-numeracy country groups (hypothesis H4a is rejected). More importantly, the difference in average performance between high-performing and low-performing countries is larger than the within-country group difference in performance between non-disabled and disabled children categories (hypothesis H4b is supported). Disabled children in the high numeracy skill countries perform even better than the non-disabled children in the low numeracy skill countries.

Except for children with multiple disabilities characterised by low enrolment and low numeracy skill returns to completed school years, the main challenge for most children with disabilities is the low school enrolment. This is especially the case for children in low-numeracy skill countries. This suggests that the priority for the education policy in low-income African countries is to improve children's school enrolment, especially for children with disabilities. The fact that the within-group differences between children with and without disabilities are similar between the low- and high-numeracy skill country groups suggests that disabled children benefit equally when the school quality improves. It demonstrates substantial room for improvement in the school system, and such enhancements also benefit disabled children. Disability effects in numeracy skills across country groups are more fundamental than the within-group gaps. Therefore, improving overall school quality and promoting school attendance for disabled children are crucial for better school performance among disabled children in the African context.

We acknowledge several limitations of this study. First, the study is limited to numeracy skills and may not be correlated with other benefits from schooling. Second, the study assesses fundamental numeracy skills and may have failed to capture substantial variation in more advanced numeracy skills that may vary more, especially among the older children that the test can identify. The test may be more appropriate in low-numeracy countries and for younger children. In high-numeracy countries, about 40% of all children answered over 90% of the questions correctly. It might lead to a potentially underestimated disability effect in high-numeracy countries. Third, while the MICS surveys are nationally representative samples aiming to provide data on the general population of children, the incidence of severe disability is very low in the population. Therefore, we have merged some categories to achieve sufficient sample sizes for statistical analysis. Last, the eight African countries included in this study are the countries that recently conducted the sixth round of the MICS survey. We do not know the external validity of the conclusions drawn based on the data from these eight countries.

For future research, we recommend studies with a broader range of skills, such as reading and science skills, and tests with more advanced numeracy skills for older children. Covering more countries with large samples may also be possible to do more statistical analyses for more disaggregated samples with rare forms of disabilities.

## Supporting information

**S1 Table. The sample size of children who have done the numeracy test by disability status and country.**
(PDF)

**S2 Table. Regression results for estimating the determinant factors of each disability type.**
(PDF)

**S3 Table. Sample characteristics.**
(PDF)

**S4 Table. Regressions on the mean numeracy score with age dummy by countries (non-disabled children).**
(PDF)

**S5 Table. Regression on the completed school years.**
(PDF)

## Acknowledgments

This paper has been undertaken as part of the research project "Education outcome variability in children with disabilities: Structure, institution or agency?". Valuable comments were received from the project coordinator Anne Hatløy and other project partners and the participants in the 44th Annual Meeting of the Norwegian Association of Economists, September 25–26, 2022. The authors take full responsibility for any remaining errors.

## Author Contributions

**Conceptualization:** Huafeng Zhang, Stein T. Holden.

**Data curation:** Huafeng Zhang.

**Formal analysis:** Huafeng Zhang, Stein T. Holden.

**Methodology:** Huafeng Zhang, Stein T. Holden.

**Software:** Huafeng Zhang.

**Supervision:** Stein T. Holden.

**Writing – original draft:** Huafeng Zhang.

**Writing – review & editing:** Huafeng Zhang, Stein T. Holden.

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
