## [Decision Letter · Decision Letter 0]

8 Mar 2023

PONE-D-22-30702Numeracy skills learning of children in Africa: - Are disabled children lagging behind?PLOS ONE

Dear Dr. Zhang,

Thank you for submitting your manuscript to PLOS ONE. After careful consideration, we feel that it has merit but does not fully meet PLOS ONE’s publication criteria as it currently stands. Therefore, we invite you to submit a revised version of the manuscript that addresses the points raised during the review process.

Major Revisions Required

We look forward to receiving your revised manuscript.

Kind regards,

Verda Salman, PhD

Academic Editor

PLOS ONE

Journal Requirements:

This paper has been undertaken as part of the research project “Education outcome variability in children with disabilities: Structure, institution or agency?” funded by the Research Council of Norway (Project Number 300635). 

However, funding information should not appear in the Acknowledgments section or other areas of your manuscript. We will only publish funding information present in the Funding Statement section of the online submission form. 

The paper has been undertaken as part of the research project “Education outcome variability in children with disabilities: Structure, institution or agency?” funded by the Research Council of Norway (Project Number 300635). 

Additional Editor Comments:

Major revisions required.

Reviewers' comments:

Reviewer's Responses to Questions

**Comments to the Author**

1. Is the manuscript technically sound, and do the data support the conclusions?

Reviewer #1: Yes

Reviewer #2: Yes

2. Has the statistical analysis been performed appropriately and rigorously? 

Reviewer #1: Yes

Reviewer #2: Yes

3. Have the authors made all data underlying the findings in their manuscript fully available?

Reviewer #1: Yes

Reviewer #2: Yes

4. Is the manuscript presented in an intelligible fashion and written in standard English?

Reviewer #1: Yes

Reviewer #2: Yes

5. Review Comments to the Author

Reviewer #1: I have reviewed the research paper “ Numeracy skills learning of children in Africa: - Are disabled children lagging behind?”. I completed the assessment as per the shared guidelines and enclosed the following observations.

1. The literature review needs to be more extensive and more extensive in scope. Please revise your literature review; the methodology part needs to be added to most of the papers you mentioned( Like which methodology they used).

2. At the end of section 3, Please explain why you have used this methodology if there is any other methodology in the previous literature, the pros and cons of using the current methodology, and why it is superior to other methodologies in literature.

3. Consider the conclusion section seriously; it is not up to the mark in its present form. Please divide conclusions into three parts, as if you are telling someone about your research findings in 5 to 7 lines. Then based on your results, discuss the policy implications, limitations, and the way forward for future researchers.

Reviewer #2: The study includes a thorough analysis and addresses a relevant gap in the literature. The following may, however, be taken into consideration as a research article:

1. The manuscript is not in conformity with the journal's style and format (reference style, table specification, etc.). (Perhaps arrangements have been made with the journal editors.)

2. Writing concisely in this study with several sections would be preferred by readers. The introduction and conceptual framework section, for instance, could be written very concisely. It is also possible to discuss the "MICS survey" only in the methodology section.

3. "The response rates to the numeracy test among different groups of children are generally quite high (about 95% or higher) but much lower among the never-in-school disabled sample (76.1%)". (If by 95% or higher you are referring to EISC, for clarity's sake, it may be indicated appropriately or better yet, reframed.)

4. Please indicate the limitations of the study (aside from the fact that the test was able to evaluate only basic numeracy skills). For instance, were all the assumptions you mentioned met? A separate section for outlining such limitations and directions for future research would be appreciated.

6. PLOS authors have the option to publish the peer review history of their article (what does this mean?). If published, this will include your full peer review and any attached files.

Reviewer #1: No

Reviewer #2: **Yes: **Francis Acquah

---

## [Author Response · Author response to Decision Letter 0]

29 Mar 2023

Responses to editor:

Response:

The reference style and format are revised to meet PLOS ONE’s style requirements.

This paper has been undertaken as part of the research project “Education outcome variability in children with disabilities: Structure, institution or agency?” funded by the Research Council of Norway (Project Number 300635). 

However, funding information should not appear in the Acknowledgments section or other areas of your manuscript. We will only publish funding information present in the Funding Statement section of the online submission form. 

Please remove any funding-related text from the manuscript and let us know how you would like to update your Funding Statement.

Response:

The funding resource is removed from the manuscript. We confirm that the funding statement does not need revision.

Responses to reviewers:

Reviewer #1: 

I have reviewed the research paper "Numeracy skills learning of children in Africa: - Are disabled children lagging behind?". I completed the assessment as per the shared guidelines and enclosed the following observations.

Response:

We would like to thank the reviewer for the valuable comments that have helped us to improve upon the paper.

1. The literature review needs to be more extensive and more extensive in scope. Please revise your literature review; the methodology part needs to be added to most of the papers you mentioned (Like which methodology they used).

Response: 

The reviewer was not specific on his/her suggested scope extension in the literature review. Therefore, we revise the literature review to include several aspects which we think are most relevant to this paper. We first discuss the empirical evidence of disabled children's education in comparable studies, mainly available w.r.t. school attendance and enrolment but not school performance/achievement. We then present several studies in developing countries assessing the school performance of disabled children. But they do not differentiate different disability types. Finally, we discuss a few studies that differentiate disability types, mainly available in Asian countries.

A brief description of the methodology used by those studies is now included in section 3.

2. At the end of section 3, Please explain why you have used this methodology, if there is any other methodology in the previous literature, the pros and cons of using the current methodology, and why it is superior to other methodologies in literature.

Response: 

In section 3, before discussing the methodology used by this paper, the methodology applied by previous studies is discussed. Then we discuss the methodology used in this paper. We suggest considering children's completed school years as both the outcome of disability and as an endogenous treatment of skill learning. Therefore, we propose to use the instrumental variable method to control for the potential endogeneity bias associated with completed school years.

3. Consider the conclusion section seriously; it is not up to the mark in its present form. Please divide conclusions into three parts, as if you are telling someone about your research findings in 5 to 7 lines. Then based on your results, discuss the policy implications, limitations, and the way forward for future researchers.

Response: 

The conclusion section is rewritten to include the following:

Research findings

Policy implications and limitations

The way forward for future research

 

Reviewer #2: 

The study includes a thorough analysis and addresses a relevant gap in the literature. The following may, however, be taken into consideration as a research article:

1. The manuscript is not in conformity with the journal's style and format (reference style, table specification, etc.). (Perhaps arrangements have been made with the journal editors.)

Response: 

We would like to thank the reviewer for the constructive assessment and comments. 

The reference style and format are revised.

2. Writing concisely in this study with several sections would be preferred by readers. The introduction and conceptual framework section, for instance, could be written very concisely. It is also possible to discuss the "MICS survey" only in the methodology section.

Response: 

The introduction and conceptual framework section are revised to be more concise. The "MICS survey" discussions are merged to the methodology section and deleted from the introduction section.

3. "The response rates to the numeracy test among different groups of children are generally quite high (about 95% or higher) but much lower among the never-in-school disabled sample (76.1%)". (If by 95% or higher you are referring to EISC, for clarity's sake, it may be indicated appropriately or better yet, reframed.)

Response: 

The sentence is revised to clarify the group of children the number refers to.

4. Please indicate the limitations of the study (aside from the fact that the test was able to evaluate only basic numeracy skills). For instance, were all the assumptions you mentioned met? A separate section for outlining such limitations and directions for future research would be appreciated.

Response: 

The conclusion section is rewritten to include the following:

Research findings

Policy implications and limitations

The way forward for future research

---

## [Editor Report · Decision Letter 1]

10 Apr 2023

Numeracy skills learning of children in Africa: - Are disabled children lagging behind?

PONE-D-22-30702R1

Dear Dr. Zhang,

We’re pleased to inform you that your manuscript has been judged scientifically suitable for publication and will be formally accepted for publication once it meets all outstanding technical requirements.

Kind regards,

Verda Salman, PhD

Academic Editor

PLOS ONE

Additional Editor Comments (optional):

Accepted for publication
---

## [Editor Report · Acceptance letter]

13 Apr 2023

PONE-D-22-30702R1 

Numeracy skills learning of children in Africa: - Are disabled children lagging behind? 

Dear Dr. Zhang:

I'm pleased to inform you that your manuscript has been deemed suitable for publication in PLOS ONE. Congratulations! Your manuscript is now with our production department. 

Kind regards, 

on behalf of

Dr. Verda Salman 

Academic Editor

PLOS ONE